# The Management of Asymptomatic Congenital Pulmonary Airway Malformation: Results of a European Delphi Survey

**DOI:** 10.3390/children9081153

**Published:** 2022-07-30

**Authors:** Casper M. Kersten, Sergei M. Hermelijn, Dhanya Mullassery, Nagarajan Muthialu, Nazan Cobanoglu, Silvia Gartner, Pietro Bagolan, Carmen Mesas Burgos, Alberto Sgrò, Stijn Heyman, Holger Till, Janne Suominen, Maarten Schurink, Liesbeth Desender, Paul Losty, Henri Steyaert, Suzanne Terheggen-Lagro, Martin Metzelder, Arnaud Bonnard, Rony Sfeir, Michael Singh, Iain Yardley, Noor R. V. M. Rikkers-Mutsaerts, Cornelis K. van der Ent, Niels Qvist, Des W. Cox, Robert Peters, Michiel A. G. E. Bannier, Lucas Wessel, Marijke Proesmans, Michael Stanton, Edward Hannon, Marco Zampoli, Francesco Morini, Harm A. W. M. Tiddens, René M. H. Wijnen, Johannes M. Schnater

**Affiliations:** 1Department of Pediatric Surgery, Erasmus MC Sophia Children’s Hospital, 3015 GD Rotterdam, The Netherlands; c.m.kersten@erasmusmc.nl (C.M.K.); s.hermelijn@erasmusmc.nl (S.M.H.); r.wijnen@erasmusmc.nl (R.M.H.W.); 2Department of Pediatric Surgery, Great Ormond Street Hospital, London WC1N 3JH, UK; dhanya.mullassery@gosh.nhs.uk (D.M.); nagarajan.muthialu@gosh.nhs.uk (N.M.); 3Department of Pediatric Pulmonology, Ankara University School of Medicine, 06620 Ankara, Turkey; drncobanoglu@yahoo.com; 4Department of Pediatric Pulmonology and Cystic Fibrosis, Hospital Universitari Vall d’Hebron, 08035 Barcelona, Spain; silvia.gartner@vallhebron.cat; 5Department of Fetus, Newborn and Infant, Bambino Gesù Pediatric Hospital, Tor Vergata University, 00165 Rome, Italy; pietro.bagolan@opbg.net; 6Department of Pediatric Surgery, Karolinska Institutet, 171 77 Stockholm, Sweden; carmen.mesas.burgos@ki.se; 7Department of Pediatric Surgery, University Hospital Padua, 35128 Padua, Italy; alberto.sgro@aopd.veneto.it; 8Department of Pediatric Surgery, ZNA-GZA Children’s Hospital, 2020 Antwerp, Belgium; stijn.heyman@zna.be; 9Department of Pediatric Surgery, Medical University of Graz, 8036 Graz, Austria; holger.till@medunigraz.at; 10Department of Pediatric Surgery, Helsinki Children’s Hospital, 00290 Helsinki, Finland; janne.suominen@hus.fi; 11Department of Pediatric Surgery, Radboud University Medical Centre, Amalia Children’s Hospital, 3525 GA Nijmegen, The Netherlands; maarten.schurink@radboudumc.nl; 12Department of Pediatric Surgery, Ghent University Hospital, 9000 Ghent, Belgium; liesbeth.desender@ugent.be; 13Department of Pediatric Surgery, Institute of Life Course and Medical Sciences, University of Liverpool, Liverpool L69 72B, UK; paul.losty@liverpool.ac.uk; 14Department of Pediatric Surgery, Queen Fabiola Children’s University Hospital, 1020 Brussels, Belgium; henri.steyaert@huderf.be; 15Department of Pediatric Pulmonology, Emma Children’s Hospital Amsterdam University Medical Centers, 1105 AZ Amsterdam, The Netherlands; s.w.terheggenlagro@amsterdamumc.nl; 16Department of Pediatric Surgery, Medizinische Universität Wien, 1090 Vienna, Austria; martin.metzelder@meduniwien.ac.at; 17Department of Pediatric Surgery, Hôpital Universitaire Robert Debré, 75019 Paris, France; arnaud.bonnard@aphp.fr; 18Department of Pediatric Surgery, Hôpital Jeanne de Flandre, 59000 Lille, France; rony.sfeir@chru-lille.fr; 19Department of Pediatric Surgery, Birmingham Children’s Hospital, Birmingham B4 6NH, UK; michael.singh2@nhs.net; 20Department of Pediatric Surgery, Evelina London Children’s Hospital, London SE1 7EH, UK; iain.yardley@gstt.nhs.uk; 21Department of Pediatric Pulmonology, Leiden University Medical Centre, 2033 ZA Leiden, The Netherlands; e.r.v.m.rikkers-mutsaerts@lumc.nl; 22Department of Pediatric Pulmonology, University Medical Centre Utrecht, 3584 CX Utrecht, The Netherlands; k.vanderent@umcutrecht.nl; 23Department of Pediatric Surgery, Odense University Hospital, 5000 Odense, Denmark; niels.qvist@rsyd.dk; 24Department of Pediatric Pulmonology, Children’s Health Ireland at Crumlin, D12 N512 Dublin, Ireland; des.cox@olchc.ie; 25Department of Pediatric Surgery, Royal Manchester Children’s Hospital, Manchester M13 9WL, UK; r.peters@nhs.net; 26Department of Pediatric Pulmonology, Maastricht University Medical Centre, 6229 HX Maastricht, The Netherlands; michiel.bannier@mumc.nl; 27Department of Pediatric Surgery, University Hospital Mannheim, 68167 Mannheim, Germany; lucas.wessel@medma.uni-heidelberg.de; 28Department of Pediatric Pulmonology, University Hospital Leuven, 3000 Leuven, Belgium; marijke.proesmans@uzleuven.be; 29Department of Pediatric Surgery, University Hospital Southampton, Southampton SO16 6YD, UK; michael.stanton@uhs.nhs.uk; 30Department of Pediatric Surgery, Leeds Children’s Hospital, Leeds LS1 3EX, UK; edward.hannon1@nhs.net; 31Department of Pediatric Pulmonology, Red Cross War Memorial Children’s Hospital, Cape Town 7700, South Africa; m.zampoli@uct.ac.za; 32Department of Neonatal Surgery, Azienda Ospedaliero Universitaria Meyer, 50139 Florence, Italy; francesco.morini@meyer.it; 33Department of Pediatric Pulmonology and Allergology, Department of Radiology, Erasmus MC Sophia Children’s Hospital, 3015 GD Rotterdam, The Netherlands; h.tiddens@erasmusmc.nl

**Keywords:** congenital lung abnormalities, congenital pulmonary airway malformation, core outcome set, outcome parameters, consensus

## Abstract

Consensus on the optimal management of asymptomatic congenital pulmonary airway malformation (CPAM) is lacking, and comparison between studies remains difficult due to a large variety in outcome measures. We aimed to define a core outcome set (COS) for pediatric patients with an asymptomatic CPAM. An online, three-round Delphi survey was conducted in two stakeholder groups of specialized caregivers (surgeons and non-surgeons) in various European centers. Proposed outcome parameters were scored according to level of importance, and the final COS was established through consensus. A total of 55 participants (33 surgeons, 22 non-surgeons) from 28 centers in 13 European countries completed the three rounds and rated 43 outcome parameters. The final COS comprises seven outcome parameters: respiratory insufficiency, surgical complications, mass effect/mediastinal shift (at three time-points) and multifocal disease (at two time-points)**.** The seven outcome parameters included in the final COS reflect the diversity in priorities among this large group of European participants. However, we recommend the incorporation of these outcome parameters in the design of future studies, as they describe measurable and validated outcomes as well as the accepted age at measurement.

## 1. Introduction

The debate is ongoing on whether asymptomatic congenital pulmonary airway malformation (CPAM) patients require surgical resection versus a conservative follow-up scheme [1,2,3,4]. Those in favor of surgical resection suggest that the risk of recurrent infection or acute respiratory distress can complicate subsequent surgery [3,5]. Furthermore, CPAM lesions have been associated with malignancy, which for some warrants an elective surgical resection [6,7,8]. Those in favor of a conservative follow-up scheme emphasize that the majority of these lesions remain asymptomatic and that some may even regress spontaneously [5,9,10]. Additionally, they state that malignancy is extremely rare and that cases of malignancy have also been described following surgical resection [2,5,11,12]. The disagreement on this subject was already highlighted in 2015 by the simultaneous publication of two opposing articles in Seminars in Pediatric Surgery, one of which advocates surgical resection while the other advocates a conservative follow-up scheme in asymptomatic CPAM [2,3]. Apart from this, multiple studies reflect on the debate surrounding the management of this patient group [13,14,15].

Previous research in CPAM patients examined a wide range of outcome parameters. For example, several studies focused on symptom development in patients that were initially asymptomatic during the neonatal period. The reported proportion ranges from 3% to 64%, although follow-up duration varied from one month to several years [5,10,16,17,18]. Other studies focused on pulmonary function in CPAM patients and tested a variety of techniques, including plethysmography, spirometry and exercise tolerance testing [19,20,21,22,23]. Between these studies, the subjects’ age varied from several months to twelve years, and a comparison between those subjects that underwent surgery and those that were managed conservatively was not universally present. Further studies focused on the presence of pulmonary symptoms or reported the prevalence of certain radiological abnormalities or chest wall deformities during follow-up [10,24,25].

The abovementioned studies are valuable in understanding the natural history of patients with CPAM. However, a reliable comparison between them is complicated by the variety in outcome parameters, measurement methods and subjects’ age.

Selecting appropriate outcome parameters is often the first step towards a uniform design of studies, which in turn will facilitate comparability between interventions. The Core Outcome Measures in Effectiveness Trials (COMET) initiative was developed to guide the process of selecting appropriate outcome parameters, and has already been utilized to develop multiple core outcome sets for a variety of topics [26,27,28,29]. Such a core outcome set has not yet been developed for the management of CPAM, or congenital lung lesions in general. The primary aim of this study was therefore to investigate if defining such a core outcome set for CPAM lesions was feasible among a large group of specialized caregivers in Europe, despite the opposing standpoints between treating specialists.

## 2. Materials and Methods

This study was performed according to the Delphi consensus methods of the COMET handbook [26]. The previously published study protocol provides a detailed description of the methods [30]. In short, participants were recruited through a pre-existing network of pediatric surgeons and pediatric pulmonologists interested in collaboration concerning the management of CPAM. Potential participants were encouraged to enroll colleagues, who received an email explaining the aims and procedures of the Delphi survey. Three extensive literature reviews were scrutinized for outcome measures, which were formulated to match a standard format: (1) the outcome parameter, (2) the measurement instrument and (3) the age at assessment [2,3,5]. Participants were asked to score every outcome parameter according to the following question: “How important would you rate the following outcome parameter for determining the optimal management of asymptomatic CPAM patients?”. All included outcome parameters focused on postnatal, asymptomatic patients, regardless of the chosen management strategy and the moment during follow-up. During the survey, participants could suggest additional outcome parameters, which were added provided they were indeed original. Participants could also add new time-points for an outcome parameter. A three-round Delphi survey was performed, in which participants had 4 weeks to complete their scoring for each round and received a weekly reminder email. Participants were assigned to either a surgical stakeholder group or a non-surgical group, and were asked to score outcome parameters according to a nine-point Likert scale. This scale consisted of three categories: not important (scoring 1–3), important but not critical (scoring 4–6) and critical (scoring 7–9). Those who had completed the first round were invited to participate in the second round. They were shown their own scoring of the first round as well as the median scoring of their own stakeholder group for all outcome parameters. The instruction was to adapt their own scoring if desirable, and to score the additionally suggested outcome parameters. Preceding the third round, we excluded those outcome parameters that met the following preset exclusion criteria: >70% of the participants rated the outcome as not important (scoring 1–3) and <15% rated it as critical (scoring 7–9). All participants who had completed the second round were invited to participate in the third round, which proceeded in analogy to the second round, except that the median scores of both stakeholder groups were provided and no new outcome parameters were offered. Outcome parameters met the preset criteria for consensus when >70% of the participants rated the outcome 7–9 and <15% rated it 1–3. All outcome parameters that reached consensus were included in the final COS. Finally, all participants were asked to state in their own words which outcome parameter they perceived as the most important in determining the best management of asymptomatic CPAM patients.

### Data Analysis

Throughout the Delphi survey, the publicly accessible ‘Welphi’ survey tool was used to record input from participants [31]. Anonymized data were stored on a secure online server and managed according to the European General Data Protection Regulation [32]. Data were analyzed with the statistical package SPSS (version 25, IBM Corp., Armonk, NY, USA). Following the final survey round, the median scoring of each outcome parameter was calculated per stakeholder group and compared between the two groups using the Mann–Whitney U test. Scores for each parameter were divided in three categories: not important (scoring 1–3), important but not critical (scoring 4–6) and critical (scoring 7–9). The percentage of scoring in each category was calculated, after which the final COS was defined according to the preset criteria.

Attrition rate was assessed separately for each round and stakeholder group by calculating the percentage of non-continuing participants, in accordance with the COMET handbook guidelines [26].

To prevent an overestimation of the degree of consensus due to selective drop-out during the survey, attrition bias was tested separately in each stakeholder group for each round of the Delphi survey [33]. For each outcome parameter, the Mann–Whitney U test served to compare average scores between those only completing the round in question and those who also completed the following round.

## 3. Results

Twenty initial outcome parameters were obtained from the literature review, which were pooled into ten categories: respiratory symptoms, infection, parental anxiety, quality of life, anthropometric measurements, lesion characteristics, spirometry results, exercise tolerance, surgical complications, and malignancy. A flowchart detailing the three-round Delphi survey process is shown in Figure 1.

### 3.1. Round 1

A total of 79 potential participants (45 (57%) surgeons; 34 (43%) non-surgeons) were invited for the first round, of whom 63 (80%) participants completed this round (39 (62%) surgeons; 24 (38%) non-surgeons). A total of 20 outcome parameters were scored in the first round and participants suggested 23 additional parameters. No parameters were excluded after the first round.

### 3.2. Round 2

All 63 participants who completed the first round were invited for the second round, which was completed by 60 (95%) participants (36 (60%) surgeons; 24 (40%) non-surgeons). A total of 43 outcome parameters were scored or rescored. After the second round, no parameters met the criteria for exclusion; thus, all parameters were carried through to the third round.

### 3.3. Round 3

All 60 participants who completed the second round were invited for the third round, which was completed by 55 (92%) participants (33 (60%) surgeons; 22 (40%) non-surgeons), representing 28 centers in 13 countries. An overview of all outcome parameters, and their assigned scores, can be found in Table 1. A total of 43 outcome parameters were scored in the third round and 17 parameters across eight domains were scored significantly different between the stakeholder groups.

### 3.4. Final Core Outcome Set

Table 2 shows the final COS. A total of 7 out of the 43 (16%) outcome parameters scored during the final survey round met the criteria for consensus. These outcome parameters were distributed across three domains: respiratory symptoms, surgical complications and lesion characteristics.

Eight outcome parameters met the criteria for consensus in only one of the stakeholder groups, and were therefore not included in the final COS (Table 3). Within the surgical stakeholder group, the following outcome parameters met the criteria for consensus: histology proven malignancy, fever and consolidation on X-ray at the site of the CPAM, and hospital admission with intravenous (IV) antibiotics due to lower respiratory tract infection (LRTI). Within the non-surgical stakeholder group, consensus was achieved on the outcome parameters wheezing (defined as >3 episodes per year), multiple spirometry measurements (FVC, FEF_25–75_ and FEV_1_), and evidence of systemic blood supply to the lesion. After completing the final round of the survey, participants were asked to choose one outcome parameter that they found to be the most important for determining the optimal management for asymptomatic CPAM patients, the results of which can be found in Table 4.

### 3.5. Attrition Rate and Attrition Bias

Thirty-three out of the 39 surgeons who completed the first round completed all three rounds, resulting in an attrition rate of 15%. The corresponding figures for the non-surgeons are 22 out of 24, resulting in an attrition rate of 8%. No significant attrition bias was found in either stakeholder group for any of the rounds of the Delphi survey.

## 4. Discussion

This study aimed to reach consensus on a COS for the management of asymptomatic CPAM patients among a large group of specialized caregivers across Europe. Universal use of standardized outcome parameters in a COS has the potential to increase the quality and relevance of future research in asymptomatic CPAM patients. The online, three-round Delphi survey was completed in 28 specialized medical centers across Europe, which led to consensus for seven outcome parameters: respiratory insufficiency, surgical complications, mass effect/mediastinal shift (at three time-points) and multifocal disease (at two time-points).

An important outcome parameter was respiratory insufficiency requiring supplemental oxygen, ventilation and/or surgical resection, which is unsurprising as this was found to be a major reason to resect a CPAM in a recent European survey [1]. Nevertheless, retrospective studies—with generally limited follow-up—estimated that only 3–24% of initially asymptomatic patients become symptomatic through infancy [5,10,16,34,35]. Long-term follow-up studies in asymptomatic CPAM patients are lacking, preventing accurate estimation of long-term symptom development rate in this patient group. Future research with long-term follow-up is therefore urgently needed. On the other side of symptom development, surgical complications are also included in this COS. Again, a longitudinal comparison between surgical complication rates and the potential burden of conservative follow-up is lacking in literature.

This COS includes radiological abnormalities, namely mediastinal shift and multifocal disease at various ages. In this respect, previous studies have only investigated the correlation between prenatally diagnosed mediastinal shift and perinatal outcome, showing ambiguous results, possibly resulting from the various measurement methods [36,37,38,39]. The relation between mediastinal shift and long-term outcome is yet to be investigated. Somewhat surprisingly, the COS produced in this study does not include outcome parameters directly linked to lesion size, even though several studies found a correlation between lesion size and symptom development [40,41]. However, lesion size can be linked to mediastinal shift to a certain degree, as a large lesion will have a higher chance of causing this shift. Multifocal congenital pulmonary disease, also included in this COS, has been described in several studies in association with a broad differential diagnosis, including bufllae, pneumatoceles, infectious processes, and malignancies such as pleuropulmonary blastoma (PPB) and adenocarcinoma in situ (AIS, formerly known as bronchoalveolar carcinomar (BAC)) [42,43]. The relationship between multifocal disease and the development of symptoms as well as outcome in patients with CPAM remains unclear until now. The relationship between CPAM and malignancy needs further investigation because the exact incidence remains unknown and research of relevant molecular and genetic factors is ongoing [5]. However, particular caution is advised in case the lesion is diagnosed after birth, or in relatively rare cases of CPAM type 4, as these characteristics are associated with a higher risk of malignancy [8,44,45,46].

This study was initiated because of the current heterogeneity in reported outcome parameters and their measurement methods in studies concerning CPAM patients. The final COS confirms this heterogeneity by the differences in outcome parameters between the two stakeholder groups. Outcome parameters such as infection and proven malignancy after resection reached consensus among surgeons. The non-surgeons reached consensus on outcome parameters concerning wheezing, lung function and the radiological assessment of systemic blood supply to the lesion. We hypothesize that surgical specialists consider outcome parameters linked to the indication for surgical resection in CPAM patients most important, while pediatric pulmonologists and neonatologists consider outcome parameters concerning pulmonary morbidity and radiological assessment more important, resulting in a slight bias per stakeholder group. The differences in scoring between the two stakeholder groups highlight the impact of multidisciplinary care for CPAM patients and emphasize the importance of taking all different perspectives into consideration in the design of future studies.

The outcome parameters included in the final COS do not fully overlap with those mentioned by participants as most important for determining the optimal management for asymptomatic CPAM patients. For example, while the final COS does not include infection-related outcome parameters, infection-related outcome was provided as the most important by 32% of participants, the most often of all outcome categories. The opposite was true for the outcome parameter of respiratory insufficiency, which achieved the highest level of consensus and is included in the final COS, while only three participants (6%) scored it as the most important outcome parameter. These dissimilarities, again, confirm the diversity in opinions among specialist caregivers concerning the management of asymptomatic CPAM patients.

To our knowledge, this is the first international Delphi survey on the management of asymptomatic CPAM patients. A valuable strength of this study is the large number of participants that completed the three-round survey: 55 representatives from 28 specialized centers spread across Europe. The design of this three-round survey proved to function according to the preset criteria, and attrition rate and attrition bias were both within limits. In other words, the drop-out rate during the survey was acceptable and there was no selective drop-out of participants with a certain (minority) opinion.

The lack of patient and public involvement in the development of the final COS is a limitation of this study. The most important consideration here was the knowledge that established patient societies or patient-support groups are scarce and widespread, making it challenging to establish an unbiased general opinion on the most important outcome parameters for this group [47]. Furthermore, we imagine that parents of asymptomatic patients might have difficulty providing input on desired outcome measures, as the large majority will not be confronted with symptoms or complications in their children [30].

Another limitation of this study can be found in the merging of the input from both stakeholder groups into one shared core outcome set, leading to the loss of certain outcome parameters that scored high in only one of the two stakeholder groups. However, in our opinion, this situation quite accurately reflects the daily clinical practice, in which multidisciplinary teams decide on the management of patients with congenital lung disease. Lastly, the Delphi survey was completed by more surgical specialists than non-surgical specialists (60% vs. 40%), possibly influencing the composition of the final core outcome set.

The seven outcome parameters included in this final COS can be applied for the development of prospective studies as they describe measurable and validated outcomes as well as the accepted age at measurement. For that matter, each of the outcome parameters could be included individually in a tailor-made study design. We believe that a prospective randomized study, including a long-term follow-up period, is necessary to evaluate whether surgical resection or a conservative follow-up is superior for asymptomatic CPAM patients. Furthermore, we suggest that in addition to this COS, functional outcome parameters such as endurance tests or lung function tests should be taken into consideration, as they are especially suitable for comparing outcomes in a generally asymptomatic population. Lastly, as a means to consider multiple perspectives and maximize the implication of future findings, we suggest patient and public involvement in the design of future studies in this population.

In conclusion, this study delivers the first COS intended to evaluate the management of asymptomatic CPAM patients. It reflects the diversity in priorities among the large group of European participants that contributed to the development of the COS. Consequently, the core outcome set is not a ready-to-use template for the design of future studies, though it does accurately highlight relevant individual outcome parameters. We recommend researchers to incorporate these outcome parameters in the design of future studies, as this will test the practicality—and possibly lead to the optimization—of this COS.

## Figures and Tables

**Figure 1 children-09-01153-f001:**
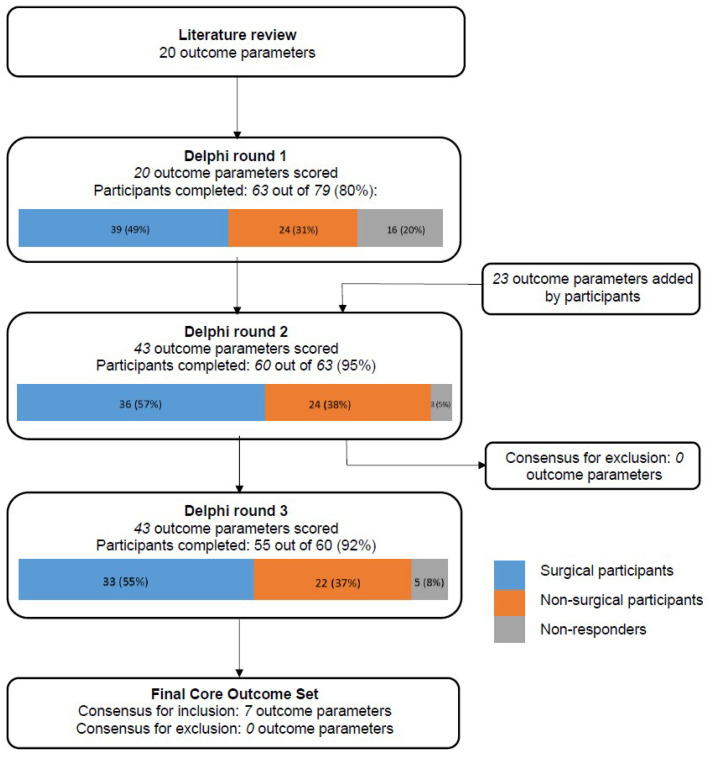
Flowchart of the 3-round Delphi Survey.

**Table 1 children-09-01153-t001:** Summary of outcome parameter scoring per stakeholder group during Delphi survey.

Outcome Category	Outcome Parameter	Age at Measurement		Median Scoring during Round 3			
				Surgical Group	Non-Surgical Group	*p*-Value	
Respiratory Symptoms	Wheezing defined as ≥3 episodes per year	All follow-up visits		4	9	<0.000	†
	Respiratory insufficiency *	All follow-up visits		9	7.5	0.006	
	Liverpool Respiratory Symptom Questionnaire (LRSQ)	All follow-up visits		6	5	0.22	
	Respiratory rate	All follow-up visits	‡	6	6	0.039	
	Reduced breath sounds on chest auscultation	All follow-up visits	‡	6	6.5	0.037	
Infection	Infection at lesion site §	All follow-up visits		9	5	<0.001	
	Parent reported hospital admission with IV antibiotics due to LRTI	All follow-up visits		7	5.5	<0.001	
	Parent reported oral antibiotics due to LRTI	All follow-up visits		6	5.5	0.2	
	Chronic inflammation in resected specimens	Not applicable	‡	6	5	0.569	
Parental Anxiety	Amsterdam Preoperative Anxiety and Information Scale (APAIS)	All follow-up visits		5	7	<0.001	
	Visual Analog Scale (VAS) scoring	All follow-up visits		5	7	<0.001	
Quality of Life	Pediatric Quality of Life Inventory (PedsQL)	All follow-up visits		6	6	0.747	
Anthropometric Measurements	Height for age measurements (WHO standards)	All follow-up visits		5	6	0.004	†
	Weight for height measurements (WHO standards)	All follow-up visits		5	6	0.025	†
Lesion Characteristics	Lesion size on first postnatal chest X-ray	<1month		6	6	0.401	
	Lesion location on CT-scan	3–9 months	‡	6	5	0.458	
	Mass effect/mediastinal shift on chest x-ray	<1 month	‡	8	8	0.757	
	Mass effect/mediastinal shift on CT-scan	3–9 months	‡	7	7	0.844	
	Mass effect/mediastinal shift on CT-scan	2.5 years	‡	7	7	0.933	
	CPAM type as seen on CT	3–9 months	‡	7	7	0.607	
	Systemic blood supply on CT-scan	3–9 months	‡	7	7	0.227	
	Multifocal disease on CT scan	3–9 months	‡	7	7	0.805	
	Multifocal disease on CT scan	2.5 years	‡	7	7	0.576	
	Quantitative lung volume measurement of first CT-scan	3–9 months		6	5	0.017	
	Quantitative lung volume measurement of follow-up CT-scan	2.5 years		6	5	0.004	
Spirometry	FEV_1_	5 years		5	7	<0.001	
	FEV_1_	10 years	‡	5	6	0.868	
	FEF_25–75_	5 years		5	9	<0.001	
	FEF_25–75_	10 years	‡	5	4.5	0.246	
	FVC	5 years		5	7	<0.001	
	FVC	10 years	‡	5	6	0.273	
	Lung clearance index	5 years	‡	4	4	0.327	
	FRC	5 years	‡	4	4	0.576	
Exercise Tolerance	Bruce treadmill test	5 years		6	6	0.157	
	VO2 max measurements	8 years	‡	5	5	0.771	
	VO2 max measurements	10 years	‡	5	5	0.804	
Surgical Complications	Surgical complications ¶	Not applicable		7	9	0.018	
Malignancy	Proven malignancy on histology by experienced pathologist	Not applicable		9	6	<0.001	
Skeletal Deformity	Physical examination	All follow-up visits	‡	6	6	0.205	
	Chest/Spine X-ray	2.5 years	‡	5	6	0.222	
	Chest/Spine X-ray	5 years	‡	6	6	0.544	
	CT-imaging	2.5 years	‡	5	6	0.299	
	CT-imaging	5 years	‡	5	6	0.392	

Significance *p* < 0.05. * Requiring supplemental oxygen ventilation and/or surgical resection. ‡ Outcome parameter added by participants during round 1. § Body temperature > 38.5 °C and consolidation on chest X-ray at the site of the CPAM. † Outcome parameter also scored significantly different in round 2. ¶ Defined as any deviation from the normal postoperative course within 30 days after surgery. IV: intravenous. LRTI: lower respiratory tract infection. FEV1: forced expiratory volume in 1 s. FVC: forced vital capacity. FEF25–75: forced expiratory flow between 25 and 75% of the vital capacity. FRC: functional residual capacity using helium dilution technique. Bruce treadmill test: maximal exercise endurance time, following standardised protocol. VO2 max: maximal oxygen usage of body during exercise, to be measured by treadmill or bicycle exercise ergometry.

**Table 2 children-09-01153-t002:** Final core outcome set.

Outcome Category	Outcome Parameter	Age at Measurement	Percentage Scoring during Round 3	
			7–9 “Critical”	1–3 “Not Important”
Respiratory Symptoms	Respiratory insufficiency *	All follow-up visits	98	0
Surgical Complications	Surgical complications ¶	Not applicable	95	2
Lesion Characteristics	Mass effect/mediastinal shift ‡	<1 month	91	3
		3–9 months	89	0
		2.5 years	89	2
	Multifocal disease on CT scan	3–9 months	73	2
		2.5 years	71	4

* Requiring supplemental oxygen, ventilation and/or surgical resection. ¶ Defined as any deviation from the normal postoperative course within 30 days after surgery. ‡ Assessed on chest X-ray/CT-scan.

**Table 3 children-09-01153-t003:** Core outcome set per stakeholder group.

Consensus	Outcome Category	Outcome Parameter	Age at Measurement	Percentage Scoring 7–9 “Critical” during Round 3	
				Surgical Group	Non-Surgical Group
Both Stakeholder Groups	Respiratory symptoms	Respiratory insufficiency *	All follow-up visits	97	100
	Surgical	Surgical complications ¶	Not applicable	91	100
	Lesion characteristics	Mass effect/mediastinal shift ‡	<1 month	94	86
			3–9 months	91	86
			2.5 years	91	86
		Multifocal disease on CT scan	3–9 months	70	77
			2.5 years	70	73
Surgical Stakeholder Group only	Malignancy	Proven malignancy on histology by experienced pathologist	Not applicable	97	14
	Infection	Infection at lesion site §	All follow-up visits	94	0
		Parent reported hospital admission with IV antibiotics due to LRTI	All follow-up visits	82	27
Non-Surgical Stakeholder Group only	Respiratory symptoms	Wheezing, defined as ≥3 episodes per year	All follow-up visits	9	100
	Spirometry	FVC	5 years	21	100
		FEF_25–75_	5 years	21	91
		FEV_1_	5 years	21	82
	Lesion characteristics	Systemic blood supply on CT-scan	3–9 months	64	77

* Requiring supplemental oxygen, ventilation and/or surgical resection. ¶ Defined as any deviation from the normal postoperative course within 30 days after surgery. ‡ Assessed on chest X-ray/CT-scan. § Body temperature > 38.5 °C and consolidation on chest X-ray at the site of the CPAM. LRTI: lower respiratory tract infection. FEV1: forced expiratory volume in 1s. FVC: forced vital capacity. FEF25–75: forced expiratory flow between 25 and 75% of the vital capacity.

**Table 4 children-09-01153-t004:** Outcome parameters scored as most important for determining the optimal management of asymptomatic CPAM patients.

Outcome Category	Outcome Parameter	Percentage Scored	
			Total
Respiratory Symptoms	Respiratory insufficiency *	6	
Infection	LRTI	16	32
	Infection at lesion site §	8	
	Frequency of antibiotics usage due to LRTI	6	
	Recurrent infection ¶	2	
Quality of Life	Quality of life	4	
Lesion Characteristics	CT-scan evaluation, dimensions of lesion	16	30
	Mass effect/mediastinal shift on CT-scan	8	
	Radiological imaging	2	
	Evaluation with expert radiologist of first CT-scan	2	
	Change in lesion size over time	2	
Spirometry	Lung function	2	4
	FEV_1_	2	
Exercise Tolerance	Symptoms or mediastinal shift on CT/x ray at 6 months of age	2	
Malignancy	Malignancy risk/suspicion	6	14
	Malignancy histologically proven	4	
	Malignancy	4	
Other	Age	2	6
	History	2	
	Oxygen supply	2	
Combinations	Symptoms or mediastinal shift on CT/x ray at 6 months of age	2	

* Requiring supplemental oxygen, ventilation and/or surgical resection. § Body temperature > 38.5 °C and consolidation on chest X-ray at the site of the CPAM. ¶ Identified by consolidation on chest X-ray at the site of the CPAM in the last year and requiring hospital admission and intravenous antibiotics. LRTI: lower respiratory tract infection. FEV1: forced expiratory volume in 1 s. Bruce treadmill test: maximal exercise endurance time, following standardised protocol.

## Data Availability

The data presented in this study are available on request from the corresponding author. The data are not publicly available due to privacy reasons.

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
