# Peer review of "The Management of Asymptomatic Congenital Pulmonary Airway Malformation: Results of a European Delphi Survey"

_children, 2022, doi:10.3390/children9081153_

Round 1
Reviewer 1 Report
I commend the authors for undertaking this very important study and for presenting it in a clear and concise manner. The core outcomes listed in the study will serve as an important tool for future studies.
Author Response
We wish to thank the reviewer for his/her assessment of our manuscript and we are happy to read the positive conclusion.
Reviewer 2 Report
1. Is this study proved by IRB ?
2. The manuscript showed that the results of outcome parameters by participants' reply, but how about the correlation between outcome parameters and clinical presentation of these CPAM patients? In other words, do all these asymptomatic CPAM not undergo operation in the futher time follow-up ? Because some of these patients present respiratory insufficiecy, and did they undergo operation ?
3. I wonder that the 5 year-old patient could perform the spirometry well ?
4. Please revised the Table 1 Exercise tolerance VO2 max measurements 8 uears --> 8 years.
5. Could author describe or discuss more detail about what's kind of infection in this study?
Author Response
We wish to thank the reviewer for his/her assessment of our manuscript, and prefer to answer the points made systematically.
1. Considering no patient information was included in this study and all data was handled anonymously, ethical approval was not indicated for this study. Nevertheless, all participants provided electronic consent for participation in the Delphi survey, as was stated in our previously published protocol manuscript on page 4 (https://bmjopen.bmj.com/content/11/4/e044544).
2. We thank the reviewer for this comment, and agree that clinical correlation should always be a priority. However, considering the complexity of the debate surrounding the management of asymptomatic CPAM, we believe the first step towards further studies should be the establishment of a core outcome set, consisting of those outcome parameters that are of interest among a large group of specialists involved in the treatment of CPAM patients. This study was therefore not designed to assess longitudinal outcome in individual CPAM patients, but rather was designed to reflect the priorities of the involved specialists in asymptomatic CPAM patients as a whole.
3. We agree with the reviewer that the spirometry results of a 5 year old patient could possibly be less trustworthy than the results of the same assessment in an older patient. The discussion on the minimal age for spirometry is still ongoing, and falls outside the scope of this study. This specific outcome parameter (spirometry in a 5 year old) was not part of the initial set of parameters, and was added by the participant following round one. To prevent censoring of outcome parameters as much as possible, and according to the COMET initiative handbook, all suggested additional outcome parameters were added to the survey, as long as they were original. This was also the case for spirometry at 5 years of age, and thus this parameter was included in the survey.
4. We apologize for this typing error, and have corrected this.
5. For this survey, 4 outcome parameters were included that considered infection: infection at lesion site (defined as T>38.5 + consolidation on chest X-ray at the site of the CPAM), parent reported hospital admission with IV antibiotics due LRTI, parent reported oral antibiotics due to LRTI, chronic inflammation in resected specimens (in case of surgical resection), please see table 1 on page 5.
Reviewer 3 Report
The management of asymptomatic Congenital Pulmonary Airway Malformation: results of a European Delphi survey
This is an interesting study, aiming to define a core outcome set (COS) for pediatric patients with an asymptomatic CPAM..
The authors found seven outcome parameters which were considered as the most important by the surgeons and non-surgeons after three rounds.
Although the study is interesting, there are several specific issues that need to be addressed.
Major comments:
The main concern is the risk of malignancy, which was not addressed and stressed enough, and I feel that it merits further discussion.
Tha authors state that “though the relationship between CPAM and the development of malignancy is still under debate”…. (lines 286-287). However, in the cited paper, it is stated clearly that “Among children with primary lung lesions initially detected after birth, PPB appears to be more common than previously thought, occurring in 10% of resected lesions. These results strongly caution against routine nonoperative management in this patient population”. Please rephrase the paragraph.
Another important paper - Congenital pulmonary airway malformations: state-of-the-art
review for pediatrician’s use, by Leblanc et al., also addresses this question, stating that “Another argument in favor of surgery is the absence of absolute concordance between radiological and pathological findings. Type 4 CPAM is indistinguishable from type I PPB at imaging”, and the authors discuss an algorythm to try and distinguish between them.
Author Response
We wish to thank the reviewer for his/her assessment of our manuscript. We have rephrased the relevant paragraph, including the references suggested by the reviewer. We agree with the reviewer on the fact that the risk of malignancy in CPAM could be discussed in more detail, even though this was not the main aim of this study. We have rephrased the paragraph, please see page 9 line 291-296.
Round 2
Reviewer 3 Report
The authors responded to the requested issues. In my opinion the paper is now acceptable for publication